# A Case Study of an 87-Year-Old Male Bodybuilder with Complex Health Conditions

**DOI:** 10.3390/medicina57070664

**Published:** 2021-06-28

**Authors:** Daniel A. Hackett, Lachlan Mitchell, Guy C. Wilson, Trinidad Valenzuela, Matthew Hollings, Maria Fiatarone Singh

**Affiliations:** 1Physical Activity, Lifestyle, Ageing and Wellbeing Faculty Research Group, Faculty of Medicine and Health, School of Health Sciences, The University of Sydney, Camperdown, NSW 2006, Australia; guy.wilson@sydney.edu.au (G.C.W.); t.valenzuela@sydney.edu.au (T.V.); matthew.hollings@sydney.edu.au (M.H.); maria.fiataronesingh@sydney.edu.au (M.F.S.); 2National Nutrition Surveillance Centre, School of Public Health, Physiotherapy and Sport Science, University College Dublin, Belfield, Dublin 4, Ireland; lachlan.mitchell@ucd.ie; 3The Hinda and Arthur Marcus Institute for Aging Research, Hebrew SeniorLife, Boston, MA 02131, USA

**Keywords:** bodybuilding, body composition, aging, resistance training, muscle strength

## Abstract

This exploratory clinical case report presents an 87-year-old man who began bodybuilding at the age of 76 years and was officially recognised as the world’s oldest competitive bodybuilder, competing until age 83. He has a background of complex health conditions including polio, strokes, cardiac arrest, atrial fibrillation, prostate disease, osteoarthritis, depression, bowel obstruction, reflux, and bladder cancer. Assessments of body composition, bone density, muscle performance, and diet-related practices were performed. The bodybuilder had superior fat-free mass, lower fat mass, and generally greater muscle performance compared to untrained healthy males of a similar age. Commencement of bodybuilding in older age appears to be possible, even with ongoing complex health conditions, and the potential benefits of this practice require systematic investigation in the future.

## 1. Introduction

Aging typically results in the deterioration of physiological function over time, greater risk of disease, and ultimately increased risk of death [1]. Muscle strength and mass will decrease 30–50% by ≈80 years [2], as accompanied by substantial reductions in bone density [3]. However, progressive loss of muscle strength and mass with aging (known as sarcopenia) [4] and bone loss can be rapidly accelerated through inactivity [5], poor dietary practices [6], and morbidity-associated catabolic factors [7]. Sarcopenia is highly prevalent among older adults with cardiovascular disease [8], prostate cancer [9], and middle-aged and older stroke survivors [10]. Additionally, the majority of poliomyelitis (polio) survivors (60–90%) do not recover full muscle strength due to varying degrees of paralysis, or may develop post-polio syndrome decades after recovery from the illness [11]. These body composition and muscle function declines lead to an increased risk of adverse events such as falls, fractures, frailty, disability, and mortality [4].

Evaluations of body composition including fat-free mass index (FFMI) and fat mass index (FMI) have been used to monitor muscularity and adiposity, respectively [12]. Briefly, FFMI is calculated via fat-free mass (FFM) divided by height^2^ (kg/m^2^), and FMI is calculated via fat mass (FM) divided by height^2^ (kg/m^2^). Older compared to younger healthy Caucasian adults generally have an increased FMI, while surprisingly FFMI appears to remain relatively stable in men across the lifespan when FFM and FM is derived from bioelectrical impediance (BIA) [12]. Briefly, FFM derived from BIA has been validated against dual energy X-ray absorptiometry (DXA) in healthy adults aged 18–94 y [13]. A more common method used to define sarcopenia via DXA is appendicular lean mass index (ALMI), which is calculated via the sum of lean mass in the upper and lower limbs divided by height (m) squared. Although, it is recognised that this method overestimates muscle mass when compared to criterion methods such as multiple-slice magnetic resonance imaging or creatinine excretion [4,14]. For comparison of muscle performance (i.e., strength or power) the normalisation of data is strongly encouraged to remove the influence of body mass, with the allometric normalisation method preferred [15]. Muscle performance can also be expressed per unit of muscle mass (known as muscle quality—MQ) to evaluate the muscle function of older adults [16].

Resistance training is effective for older adults to counteract losses of muscle strength and FFM (i.e., muscle and bone) due to the effects of aging and chronic disease [17,18]. To maximise these effects, older adults require adequate protein intakes (≈1.6 g/kg/day) and vitamin D levels (≥50 nmol/L) [19,20]. Bodybuilding consists of high-intensity resistance training and dietary manipulation to maximise FFM [21], while minimising increases in FM. For older adults, bodybuilding may counteract age-related catabolism and promote better outcomes when managing chronic and complex health conditions. This exploratory clinical case report presents the body composition, bone density, muscle performance, and diet-related practices of an 87-year-old male bodybuilder (BB-87y) with a complex medical history. Comparisons were made between BB-87y and two untrained (UT) older healthy males of a similar age. The quantification and qualification of potential benefits from bodybuilding in an older adult in this exploratory clinical case report may assist with a systematic investigation of this topic in the future.

## 2. Case

BB-87y commenced bodybuilding at 76 years of age in an attempt to improve his health. Prior to commencing bodybuilding, he lived a sedentary lifestyle, smoked tobacco, and consumed alcohol on a regular basis. He was diagnosed with polio at 21 years, suffered a minor stroke at 64 years, and required procedures for prostate disease and a twisted bowel at 65 years. At 66 years, during heart surgery for an aortic valve replacement, he suffered a cardiac arrest, which resulted in the implantation of a pacemaker. He was awarded a Guinness World Record in 2009 (age 79 years) as the oldest competitive male bodybuilder with his last competition at the age of 83 years. He competed in 11 bodybuilding competitions, and following commencing bodybuilding, suffered a minor stroke at 79 years, muscle hernia of the leg at 81 years, was diagnosed with bladder cancer at 82 years, and has had ongoing shoulder function limitations due to a rotator cuff tear that occurred around 82 years. Regular medications from age 83 years included omeprazaole (proton pump inhibitor for gastro-oseophaegal reflux), tiotropium bromide (bronchodilator for emphysema), irbesartan (beta-blocker for hypertension), digoxin (atrial fibrillation), warfarin (anticoagulant for cerebrovascular disease), and celecoxib (anti-inflammatory for osteoarthritis). The daily supplements used by BB-87y included a multi-vitamin tablet, vitamin D (1000 IU), and fish oil (2000 mg). He trained approximately 5 days per week with the duration of sessions being approximately 1 h. His resistance training involved a split routine performing 1–3 sets of 10–20 repetitions for exercises, and challenging loads (not causing volitional muscle fatigue). Aerobic exercise was performed each session and involved walking on a treadmill for 20 min (≈5.5 km/h).

For this case study, muscle strength, muscle power, and body composition results of BB-87y (163 cm, 60.7 kg) were compared to an UT (no history of resistance or high intensity aerobic training) male of 82 years (UT-82y; 164.8 cm, 66.6 kg) and 87 years (UT-87y; 174.6 cm, 77.4 kg). The data for these participants were collected from another study being conducted by the research group focused on older adults. Therefore, the two UT participants came from a convenience sample, and the selection of the UT participants for this case report was done retrospectively. The selection criteria for the UT participants included being within 5 years of BB-87y and with no history of resistance training. From the previous study dataset, the two UT participants selected were the closest in age to BB-87 (i.e., others were younger than 80 years). The physical activity of the UT males included playing golf twice per week (approximately 12 km walked) with an additional ≈2 km of walking on three days per week for UT-82y and ≈2.5 km walking every day for UT-87y. The medical history of UT-82y included atrial fibrillation (pacemaker implanted at 68 years), coronary artery disease with cardiac stent (at 67 years), hypertension, sleep apnoea, gastro-oesophageal reflux disease, inguinal hernia (repaired at 68 years), and right patellofemoral pain. Medications taken by UT-82y included warfarin, sotalol (beta-blocker), rosuvastatin (statin), pantoprazole (proton pump inhibitor), dutasteride (5-alpha reductase inhibitor), and tamsulosin (alpha1a-adrenoreceptor antagonist). The medical history of UT-87y included benign prostatic hypertrophy, inguinal hernia (repaired at 82 years), and plantar fasciitis, with no medications reported.

Tests performed in this study are listed below, with protocols previously described [22]. Muscle strength (via one-repetition maximum—1RM) and peak power (PP) were assessed using the chest press and recumbent leg press, Keiser A420 pneumatic resistance training equipment (Keiser Sports Health Equipment, Inc., Fresno, CA, USA). It was identified (via physician screening) that UT-82y had probable right patellofemoral osteoarthritis (although no pain during examination), and therefore it was decided that the leg press power test would not be performed by this participant. Values for 1RM and PP were expressed relative to body mass to the 2/3 power (kg-0.67) [15]. Upper body muscle quality (MQ) was calculated by chest press 1RM divided by arms and trunk lean mass (N/kg). Lower body MQ was calculated by leg press 1RM divided by lower body lean mass (N/kg). The UT participants were involved in two chest press and leg press 1RM sessions to reduce the risk of lack of familiarity with the tests influencing performance. The best 1RM result of the UT participants was recorded and PP testing for the exercises was only performed in the second testing session.

Body composition was assessed fasting via a whole-body dual energy X-ray absorptiometry scanner, (Lunar Prodigy, GE Medical Systems, Madison, WI, USA). Both FFMI and FMI were calculated (using the equations reported earlier). FFMI and FMI of BB-87y and UT older males were compared to percentiles on the basis of findings from Schutz et al. [12]. Serum 25-hydroxy-vitamin D was assessed via analysis of blood samples during January (estimated solar irradiance = 1089 W/m^2^), April (estimated solar irradiance = 993 W/m^2^), and June (estimated solar irradiance = 916 W/m^2^) for UT-82y, BB-87y, and UT-87y, respectively. BB-87y documented all his food, fluid, and supplements consumed over a 7-day period using a diary, and this was analysed using the FoodWorks program (Version 8; Xyris Software, Brisbane, Australia).

On the basis of the reference values for FFMI in Caucasian males >75 years [12], BB-87y was at the 90th percentile and the UT older males were at the 5th percentile (Figure 1). The FMI was at the 5th percentile for BB-87y (9.3% body fat) compared to the 75th percentile for UT-82y (33.1% body fat) and 90th percentile for UT-87y (37.6% body fat) in terms of males >75 years (Figure 1). BB-87y had superior muscle mass and substantially less fat mass compared to the UT older males. Appendicular lean mass index (ALMI) was calculated as the sum of lean mass in the upper and lower limbs divided by height (m) squared. The ALMI cut-off for low muscle mass threshold was <7 kg/m^2^ [4]. The ALMI of BB-87y (8.1 kg/m^2^) was above the low muscle mass threshold, compared to being at (UT-82y: 7 kg/m^2^) or below (UT-87y: 6 kg/m^2^) this threshold for the UT older males.

For bone mineral density (BMD), BB-87y had osteoporosis (T score < −2.5) on the femoral neck (T score = −3.1) and osteopenia (T score −2.5 to −1) of the hips and lumbar spine (T score = < −1.5) (Table 1) [23]. In comparison UT-87y had osteopenia of the femoral neck and right hip (T score = < −1.8) while UT-82y had ideal BMD (T score > −1) [23]. Serum 25-hydroxy-vitamin D was within the normal range (50–140 nmol/L) for BB-87y (106 nmol/L), UT-82y (74 nmol/L), and UT-87y (61 nmol/L). The diet analysis showed that BB-87y had an average daily energy intake of 9344 kJ (macronutrient distribution of 36.1% carbohydrates, 47.7% fats, and 16.2% protein), and his protein intake represented 1.47 g/kg/day.

Chest press and leg press 1RM and PP was expressed with allometric scaling and calculated via 1RM load or PP (W) divided by scaled body mass (kg^−0.67^). All of the 1RM performances were greater for BB-87y compared to UT older males (ranging from 13.6% to 55.8%) with a trend towards increased differences in performances when compared to UT-87y (Table 2). Chest press PP for BB-87y was slightly lower compared to UT-82y (−5%) but greater compared to UT-87y (+20.6%). Leg press PP was also greater for BB-87y compared to UT-87y (+30.8%), but no comparison could be made to UT-82y since he did not perform this measure. Chest press and leg press 1RM performances were also expressed relative to upper and lower body lean mass, respectively, to provide an indication of MQ. Upper body MQ was greater for BB-87y compared to the UT older males, while lower body MQ for BB-87y was greater than UT-87y but not UT-82y (−1.1%).

## 3. Discussion

This exploratory clinical case report presents novel insights into an older male who commenced bodybuilding at the age of 76 years following a history of complex health conditions and his continuation of this lifestyle despite ongoing health problems. On the basis of normative data of Caucasian males aged >75 years, we found that BB-87y was in the top 10% for FFMI and FMI, which was superior to the UT males who were in the bottom 25% for these indices [12]. However, despite BB-87y not showing any signs of sarcopenia, confirmed with a normal ALMI, he had poorer bone quality compared to the UT males in terms of his lower BMD. Resistance training is one of several preventive and therapeutic interventions for counteracting the loss of bone and muscle [24]. However, the effectiveness is dependent on using high mechanical loads (≈80% 1RM) and performing exercises that target large muscles that cross the hip and spine such as squats and deadlifts [25]. It is possible that BB-87y had not been following these resistance training recommendations for improving bone health. Since BB-87y had a history of low body fat following diet practices conducive for fat loss, this also may have negatively impacted his bone health [26]. No data were available to determine the body mass index (BMI) of BB-87y prior to commencing training. If he had a low BMI before bodybuilding, this could explain the low BMD observed [27]. Additionally, he had a long history of smoking, excess alcohol intake, and used certain medications (e.g., anticoagulants) and treatments for his previous bladder cancer, all of which would have negatively impacted his bone health.

Despite BB-87y having osteoporosis, he had no history of falls or osteoporotic fractures, potentially due to his avoidance of sarcopenia, which is known to contribute significantly to fracture risk (by more than 70% compared to individuals without sarcopenia) [28]. Notably, his diet appeared to be conducive for optimising muscle mass, with his protein intake close to the recommendation of ≈1.6 g/kg/day for resistance training-induced gains in FFM in healthy adults [20]. Additionally, although the participants had serum 25-hydroxy-vitamin D values within the normal ranges, measurements were taken in months with different estimated solar irradiances, which may have confounded these results.

Muscle strength (chest press and leg press 1RM) and MQ (upper and lower body) were mostly greater for BB-87y compared to the UT males. Additionally, muscle power was greater for BB-87y compared to UT-87y, although not compared to UT-82y for chest press PP. Previous research has demonstrated that upper extremity muscle power declines more rapidly than strength during aging [29]. However, it should also be noted that BB-87y had ongoing rotator cuff muscle problems for 4 years, which likely reduced the effectiveness of his training and his willingness to exert maximal force and velocity during testing, hence negatively impacting his chest press PP. Since the UT participants were less experienced with the resistance training testing procedures compared to BB-87y, their performance may not be a ‘true’ reflection of their muscle performance capacities. However, the UT participants were provided with two testing sessions to determine their 1RM with the best result being record, although the PP testing was only performed once. Therefore, it is possible that the muscle strength and power of the UT participants may have been slightly greater than documented in this case report.

## 4. Conclusions

Commencement of bodybuilding in older age appears to be possible, even with ongoing complex health conditions. However, the direct effect of bodybuilding on the health and muscle performance of BB-87y in this case study cannot be made due to the absence of data prior to him commencing bodybuilding. A future randomised controlled trial and case–control studies are warranted to enable conclusions about the efficacy and safety of bodybuilding in older adults.

## Figures and Tables

**Figure 1 medicina-57-00664-f001:**
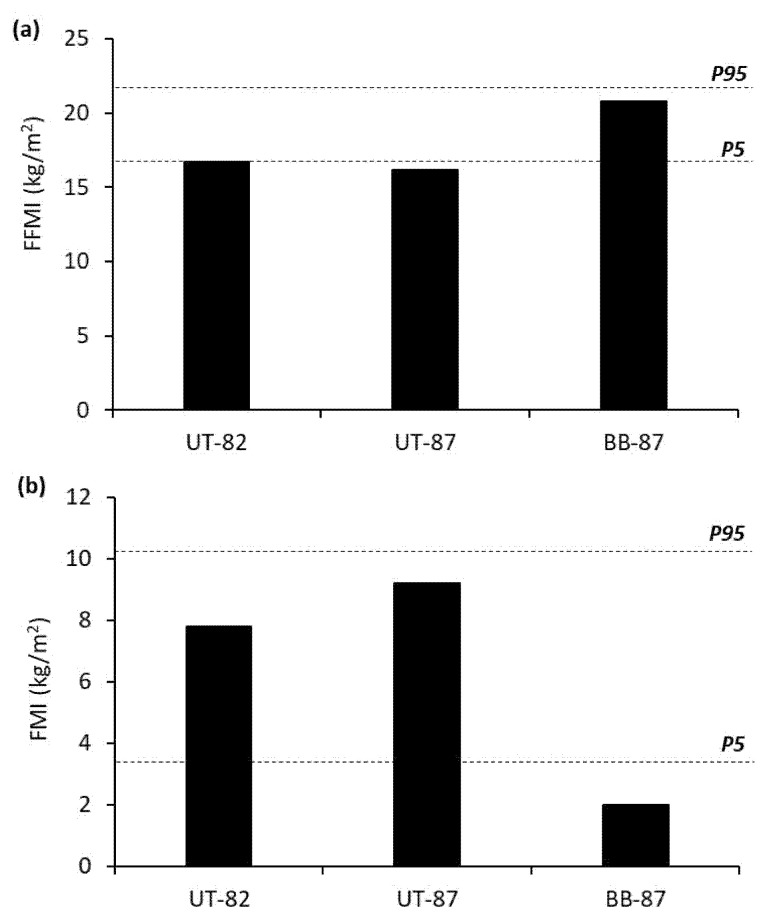
Comparison between the 87-year-old bodybuilder and untrained older males of a similar age for (**a**) fat-free mass and (**b**) fat mass indices. UT-82y = untrained male aged 82 years; UT-87y = untrained male aged 87 years; BB-87y = bodybuilder aged 87 years; FFMI = fat-free mass index; FMI = fat mass index; *P95* = 95th percentile; *P5* = 5th percentile.

**Table 1 medicina-57-00664-t001:** Differences in bone mineral density and T score for the lumbar spine, total hip, and femoral neck between the 87-year-old bodybuilder and untrained older males of a similar age.

Parameter	UT-82y	UT-87y	BB-87y	Diff BB-87y vs. UT-82y (%)	Diff BB-87y vs. UT 87y (%)
Lumbar spine (L2–L4) (g/cm^2^)	1.21	1.28	1.06	−14.2	−20.8
Right hip total (g/cm^2^)	0.97	0.84	0.77	−26	−9
Left hip total (g/cm^2^)	1	0.93	0.79	−26.6	−17.7
Right femoral neck (g/cm^2^)	0.95	0.81	0.67	−41.8	−20.9
Left femoral neck (g/cm^2^)	0.98	0.84	0.67	−46.3	−25.4
Lumbar spine T score	−0.3	0.3	−1.5	−80	−120
Right hip total T score	−0.9	−1.9	−2.4	−63	−20.8
Left hip total T score	−0.7	−1.2	−2.3	−69.6	−47.8
Right femoral neck T score	−0.9	−2	−3.1	−71	−35.5
Left femoral neck T score	−0.7	−1.8	−3.1	−77.4	−41.9

UT-82y = untrained male aged 82 years; UT-87y = untrained male aged 87 years; BB-87y = bodybuilder aged 87 years; Diff = difference. Bone mineral density was measured via a dual energy X-ray absorptiometry scanner (Lunar Prodigy, GE Medical Systems, Madison, WI, USA).

**Table 2 medicina-57-00664-t002:** Comparisons between the 87-year-old bodybuilder and untrained older males of a similar age for strength, power, and muscle quality.

Parameter	UT-82y	UT-87y	BB-87y	Diff BB-87y vs. UT-82y (%)	Diff BB-87y vs. UT-87y (%)
1RM chest press	19.2 N/kg^0.67^	13.8 N/kg^0.67^	31.2 N/kg^0.67^	+38.4	+55.8
PP chest press	14.8 W/kg^0.67^	11.2 W/kg^0.67^	14.1 W/kg^0.67^	−5	+20.6
1RM leg press	120 N/kg^0.67^	89.5 N/kg^0.67^	138.9 N/kg^0.67^	+13.6	+35.6
PP leg press	DNP	48.6 W/kg^0.67^	70.2 W/kg^0.67^	-	+30.8
Upper body MQ	12.6 N/kg	8.6 N/kg	14.2 N/kg	+11.2	+39.4
Lower body MQ	142.9 N/kg	120.2 N/kg	141.4 N/kg	−1.1	+15

UT-82y = untrained male aged 82 years; UT-87y = untrained male aged 87 years; BB-87y = bodybuilder aged 87 years; 1RM = one-repetition maximum; PP = peak power; DNP = did not perform; MQ = muscle quality; Diff = difference. Diff BB-87y versus UT calculated via [(BB-87y minus UT) divided by UT] multiplied by 100.

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
