# Peer review of "A Case Study of an 87-Year-Old Male Bodybuilder with Complex Health Conditions"

_medicina, 2021, doi:10.3390/medicina57070664_

Round 1

Reviewer 1 Report

The exploratory case report "A Case Study of an 87-year-old Male Bodybuilder with Complex Health Conditions" presents an 87-year-old man who started bodybuilding at the age of 76 and was officially recognized as the oldest competitive bodybuilder in the world, competing until the age of 83.

The findings suggest that future studies of randomized controlled trials and case-control studies should be conducted in order to draw conclusions about the efficacy and safety of bodybuilding in older adults. Furthermore, the case report concludes that initiation of weight training in older age appears to be possible even with complex ongoing health conditions.

I congratulate the authors of this case report, the subject's physiological response to previous disturbances is truly surprising. The development of future controlled studies may explain possible relationships between physical exercise and the reported variables.

Major comments:

  1. Line 93 specifies the subjects to which the case has been compared. It would be necessary to add the selection criteria for subjects UT-82 and UT-87. In case of selection sampling, what type of sampling was carried out?
  2. Line 115 states that serum Vitamin D levels were measured, can you specify what type of serum Vitamin D was measured? - You should specify the month in which the measurement was made, as well as the estimated solar irradiances for that month. If there have been measurements in months with different solar irradiances, this should be specified as it could be a confounding factor in the comparison of serum vitamin D levels.
  3. The presentation of the two Figures (a and b) contained in Figure 1 are not presented at the same height. This can be confusing and should be avoided according to "Tufte's Law".
  4. In the subject "UT-82y" described in Table 2 "DNP" appears in the "PP leg press" measurement. Please describe why the measurement was not performed. Did the right patellofemoral pain prevent the test from being performed?
  5. Lines 203 and 204 describe possible variables that impact on bone health. are there any data available to calculate body mass index prior to the start of bodybuilding? - Low body mass index prior to bodybuilding could explain the observed loss of bone mass. It should be noted in detail in this section.
  6. Other possible causes for the impact on bone health are specified in line 204-206. PLEASE include treatment with anticoagulants, treatment for previous bladder cancer, etc...
  7. In bibliographic reference 22 (line 286) reference is made to a recent "self-citation" of the manuscript "Effect of Training Phase on Physical and Physiological Parameters of Male Powerlifters". Is the "self-citation" really necessary, and are there other recent articles that can be used in which the "self-citation" can be avoided?

Minor comments:

  1. The abbreviation "UT" appears for the first time on line 95, however it is described on line 68. You must add the abbreviation the first time the term is used. Add the abbreviation on line 68.
  2. Figure 1 should follow the formatting recommendations described. If there are two graphs, (a) and (b) should be added. Please follow the procedures described.
  3. The description in Table 2 (line 183-189) is not described according to the formatting recommendations. Please make the appropriate changes.
  4. The expression of decimals is not standardized in the case report (sometimes 1 decimal place, 2 decimal places...).

Author Response

We thank Reviewer 1 for their constructive comments which have enabled us to improve the manuscript. Please find below a point-by-point response to all the comments raised. 

Comment 1: Line 93 specifies the subjects to which the case has been compared. It would be necessary to add the selection criteria for subjects UT-82 and UT-87. In case of selection sampling, what type of sampling was carried out?

Response 1: We have now provided detailed information concerning the selection of the UT participants (Please see below).

The data for these participants was collected from another study being conducted by the research group focused on older adults. Therefore, the two UT participants came from a convenience sample and the selection of the UT participants for this case report was done retrospectively. The selection criteria for the UT participants included being within 5 years of BB-87y and with no history of resistance training. From the previous study dataset the two UT participants selected were the closest in age to the BB-87 (i.e. others were younger than 80 years).”

Comment 2: Line 115 states that serum Vitamin D levels were measured, can you specify what type of serum Vitamin D was measured? - You should specify the month in which the measurement was made, as well as the estimated solar irradiances for that month. If there have been measurements in months with different solar irradiances, this should be specified as it could be a confounding factor in the comparison of serum vitamin D levels.

Response 2: The type of serum vitamin D assessed was 25-hydroxy-vitamin D and this information has now been added to the manuscript. The following has been included concerning the months of measurement:

“Serum 25-hydroxy-vitamin D was assessed via analysis of blood samples during January (estimated solar irradiance = 1089 W/m2), April (estimated solar irradiance = 993 W/m2), and June (estimated solar irradiance = 916 W/m2) for UT-82y, BB-87y, and UT-87y, respectively.”

Since there was a different estimated solar irradiance between participants when vitamin D was assessed the following has also been added.

“Additionally, although the participants had serum 25-hydroxy-vitamin D values within the normal ranges, measurements were taken in months with different estimated solar irradiances, which may have confounded these results.”

Comment 3: The presentation of the two Figures (a and b) contained in Figure 1 are not presented at the same height. This can be confusing and should be avoided according to "Tufte's Law".

Response 3: Thank you for identifying this error. The two figures are now both the same size.

Comment 4: In the subject "UT-82y" described in Table 2 "DNP" appears in the "PP leg press" measurement. Please describe why the measurement was not performed. Did the right patellofemoral pain prevent the test from being performed?

Response 4: The information below has been added to explain why this participant did not perform the leg press power test.

“It was identified (via physician screening) that UT-82y had probable right patellofemoral osteoarthritis (although no pain during examination), so it was decided that the leg press power test would not be performed by this participant.”

Comment 5: Lines 203 and 204 describe possible variables that impact on bone health. Are there any data available to calculate body mass index prior to the start of bodybuilding? - Low body mass index prior to bodybuilding could explain the observed loss of bone mass. It should be noted in detail in this section.

Response 5: The following has been added to the Discussion.

“No data was available to determine the body mass index (BMI) of BB-87y prior to com-mencing training. If he had a low BMI before bodybuilding this could also explain the low bone mass observed [27].”

Comment 6: Other possible causes for the impact on bone health are specified in line 204-206. PLEASE include treatment with anticoagulants, treatment for previous bladder cancer, etc...

Response 6: Thank you for this suggestion. The following has now been added.

“Additionally, he had a long history of smoking, and excess alcohol intake as well as the use of certain medications (e.g. anticoagulants) and treatment for previous bladder cancer which would have negatively impacted his bone health.”

Comment 7: In bibliographic reference 22 (line 286) reference is made to a recent "self-citation" of the manuscript "Effect of Training Phase on Physical and Physiological Parameters of Male Powerlifters". Is the "self-citation" really necessary, and are there other recent articles that can be used in which the "self-citation" can be avoided?

Response 7: The testing performed was very specific and to the best of our knowledge we cannot find any other studies to cite that have adequately replicated the measures used in our study. So to avoid adding in a large volume of text to describe the testing methodology used we believe that citing our previous work is justifiable.

Comment 8: The abbreviation "UT" appears for the first time on line 95, however it is described on line 68. You must add the abbreviation the first time the term is used. Add the abbreviation on line 68.

Response 8: This correction has now been made.

Comment 9: Figure 1 should follow the formatting recommendations described. If there are two graphs, (a) and (b) should be added. Please follow the procedures described.

Response 9: This has now been corrected.

Comment 10: The description in Table 2 (line 183-189) is not described according to the formatting recommendations. Please make the appropriate changes.

Response 10: Table 2 has now been amended in accordance with the formatting recommendations of the journal.

Comment 11: The expression of decimals is not standardized in the case report (sometimes 1 decimal place, 2 decimal places...).

Response 11: You are referring to the bone mineral density results which need to be expressed to 2 decimal places for the regions (to show differences rather than rounding the figures) and the T scores are only reported to 1 decimal place. We do not believe that the decimal places need to be standardized in this circumstance.

Reviewer 2 Report

This is a very impressive case report which further extends the body of evidences for resistance exercise in elderly patients. The patient has been very finely studied and compared to matched controls. Despite being a case report, this manuscript provides a convincing evidence that even very old and unhealthy individuals may have a huge benefit from exercise training. There are no special concerns to address.

Author Response

Comment 1: There are no special concerns to address.

Response 1: Thank you for taking the time to review our manuscript. Your positive feedback is very much appreciated.

Reviewer 3 Report

General comments

The authors presented health-related and muscular fitness data of an 87-year-old bodybuilder. Although the scientific significance of a case report is questionable, the paper provides interesting information and questions for future research topics. The manuscript is generally well written. Therefore, I have only some minor comments to address.

Specific comments

- line 47: Please put reference “12” in brackets.

- It can be assumed that the bodybuilder was familiar with the test procedures (chest and leg press) due to his regular training. In contrast, the strength and power test procedure were new for the untrained counterparts not.

Did the untrained persons conduct (a) familiarisation session(s)? If not, their results may be underestimated and this should be addressed in the discussion.

- Is it possible to provide information on the (health-related) quality of life of the case and the counterparts? This would add important information for the practical relevance.

Author Response

We thank Reviewer 3 for their constructive comments which have enabled us to improve the manuscript. Please find below a point-by-point response to all the comments raised. 

Comment 1: Line 47: Please put reference “12” in brackets.

Response 1: The reference has now been placed in brackets.

Comment 2: It can be assumed that the bodybuilder was familiar with the test procedures (chest and leg press) due to his regular training. In contrast, the strength and power test procedure were new for the untrained counterparts not.

Response 2: This information has been added to the Discussion as a potential confounder to the muscle strength and power results (shown below).

“Since the UT participants were less experienced with the resistance training testing procedures compared to BB-87y, their performance may not be a ‘true’ reflection of their muscle performance capacities. However, the UT participants were provided with two testing sessions to determine their 1RM with the best result being record, although the PP testing was only performed once. Therefore, it is possible that the muscle strength and power of the UT participants may be slightly greater than documented in this case report.”

Comment 3: Did the untrained persons conduct (a) familiarisation session(s)? If not, their results may be underestimated and this should be addressed in the discussion.

Response 3: The untrained participants performed two 1RM testing sessions but only one PP testing session. This information has been added to the section that describes the testing (see below).

“The UT participants were involved in two chest press and leg press 1RM sessions to reduce the risk of lack familiarity with the tests influencing performance. The best 1RM result of the UT participants was recorded and PP testing for the exercises was only performed in the second testing session.”

Comment 4: Is it possible to provide information on the (health-related) quality of life of the case and the counterparts? This would add important information for the practical relevance.

Response 4: Unfortunately no health-related quality of life measures were collected on the case (BB-87y).

Round 2

Reviewer 1 Report

The authors made corrections to the case report in line with my previous comments. The case report is well written and informative. I propose a minor revision for the case report to be proposed for acceptance:

  1. Figure 1 is presented 2 times (with change report).
  2. The description in Figure 1 (line 175-179) is not described according to the formatting recommendations (several lines of text are displayed). Please make the appropriate changes.
  3. Check line 238, as it includes a sentence that seems to continue in line 240.

I would like to take this opportunity to congratulate you on the submitted case.

Author Response

Please find below our responses to your comments.

Comment 1: Figure 1 is presented 2 times (with change report).

Response 1: We apologise for this error and have made the correction (i.e. Figure 1 now presented once)

Comment 2: The description in Figure 1 (line 175-179) is not described according to the formatting recommendations (several lines of text are displayed). Please make the appropriate changes.

Response 2: Thank you for identifying this error. We have amended the description of Figure 1 according to the journal formatting recommendations.

Comment 3: Check line 238, as it includes a sentence that seems to continue in line 240.

Response 3: Thank you for identifying this error. We have corrected so that this information is in the right order. 

Thank you again for reviewing our manuscript and providing helpful comments.

Kind regards,

Daniel Hackett